# Detailed Geogenic Radon Potential Mapping Using Geospatial Analysis of Multiple Geo-Variables—A Case Study from a High-Risk Area in SE Ireland

**DOI:** 10.3390/ijerph192315910

**Published:** 2022-11-29

**Authors:** Mirsina Mousavi Aghdam, Valentina Dentoni, Stefania Da Pelo, Quentin Crowley

**Affiliations:** 1Department of Geology, Trinity College Dublin, D02 YY50 Dublin, Ireland; 2Department of Civil and Environmental Engineering and Architecture, University of Cagliari, 09123 Cagliari, Italy; 3Department of Chemical and Geological Sciences, University of Cagliari, 09123 Cagliari, Italy

**Keywords:** geogenic radon potential, geostatistical analysis, radon-related variables, soil gas radon, airborne radiometric

## Abstract

A detailed investigation of geogenic radon potential (GRP) was carried out near Graiguenamanagh town (County Kilkenny, Ireland) by performing a spatial regression analysis on radon-related variables to evaluate the exposure of people to natural radiation (i.e., radon, thoron and gamma radiation). The study area includes an offshoot of the Caledonian Leinster Granite, which is locally intruded into Ordovician metasediments. To model radon release potential at different points, an ordinary least squared (OLS) regression model was developed in which soil gas radon (SGR) concentrations were considered as the response value. Proxy variables such as radionuclide concentrations obtained from airborne radiometric surveys, soil gas permeability, distance from major faults and a digital terrain model were used as the input predictors. ArcGIS and QGIS software together with XLSTAT statistical software were used to visualise, analyse and validate the data and models. The proposed GRP models were validated through diagnostic tests. Empirical Bayesian kriging (EBK) was used to produce the map of the spatial distribution of predicted GRP values and to estimate the prediction uncertainty. The methodology described here can be extended for larger areas and the models could be utilised to estimate the GRPs of other areas where radon-related proxy values are available.

## 1. Introduction

Radon is a radioactive gaseous element that can be found in the air, soil, rocks and water sources. The gas is produced via the decay chain of primordial radionuclides ^238^U, ^232^Th and ^235^U. The most abundant isotopes are ^222^Rn (from the decay chain of ^238^U) and ^220^Rn (from the decay chain of ^232^Th, known as thoron), which have half-lives of 3.82 days and 55.6 s, respectively. U and Th are concentrated in accessory minerals such as orthite, allanite, monazite, zircon, apatite and titanite. Such minerals occur in a variety of rock types (e.g., granite, metamorphic rocks and placer deposits) [1,2]. The main source of radon (^222^Rn) in buildings is the gas flux from soils and rocks that host uranium/radium and minerals [3]. Additional sources of radon and, particularly, thoron (thorium-bearing minerals such as the rare earth element phosphate mineral, monazite) may occur in building materials, and both gases may also be dissolved in water [4,5,6]. In the soil gas, ^220^Rn concentrations can be even higher than those of ^222^Rn [7]. Due to the shorter half-life, thoron can only migrate a short distance before it decays; therefore, building materials rather than the soil beneath the house are usually the principal source of thoron in indoor air [8]. Inhalation of radon’s solid decay products (i.e., ^218^Po, ^214^Po, ^214^Pb, ^214^Bi) can lead to the accumulation of these alpha emitters in bronchial tissues of the lung and, subsequently, increase the risk of lung cancer [9]. Thoron may decay almost completely in indoor air; however, similar to radon, a radiation dose might arise due to significant concentrations of its decay products which may still remain in situ [8]. Due to the adverse health effects of radon which are now well-known [10,11], many European countries have considered action plans to identify areas where there might be a higher radon risk [12].

Radon mapping is mainly performed in two methods: (a) based on spatial distribution of indoor radon measurements and (b) estimation of radon potential based on geogenic information such as radiometric surveys (terrestrial or airborne), geochemical data (e.g., U and Ra content), soil gas radon measurements, soil permeability, faults, etc. [9,13]. For radon maps prepared based on the spatial distribution of indoor radon concentrations at a large scale, a relatively dense measurement grid is necessary. In this case, the cost of measurements might be considerable and the produced map may only have meaning within the inhabited areas and in standard dwelling conditions (i.e., ground floor rooms, presence of a basement) [9,14]. As a result of the second approach, a geogenic radon potential map can be developed that can predict the amount of radon transported from nearby geological formations to the atmosphere, thus assisting regulators and local authorities in conducting effective land planning and designing a proper strategy to minimise radon exposure in the built environment [15]. This can be later used to simulate indoor radon levels for a standard dwelling based on a soil–indoor transfer factor [16].

In Ireland, the Environmental Protection Agency (EPA) produced a 10 km grid cell-sized radon map based on the results of indoor radon surveys in approximately 11,000 homes [17]. The accuracy and spatial resolution of the map were later improved using a logistic regression modelling technique which incorporated both indoor radon data and four separate geogenic characteristics [15]. Further refinement of the logistic regression radon map was conducted, resulting in a new map released by the EPA in May 2022. Furthermore, Elio et al. [18] utilised the Tellus airborne geophysical data and subsoil permeability to produce a geogenic radon potential map of Ireland, which does not rely on indoor radon data. In this paper, the goal is to develop a geostatistical model to predict the geogenic radon potential for a high-risk area in southeastern Ireland that is underlain by granite and metasediments. To accomplish this, airborne radioelement concentrations (i.e., eU and eTh), air-absorbed dose rate, distance from the major fault, soil permeability and a digital terrain model were considered as proxy variables (i.e., predictors). Soil gas radon concentration was also taken into account as the response value. Then, an ordinary least squared (OLS) regression model was developed and the significance of each proxy variable in supporting the proposed model together with the validity of the model was evaluated. Considering that Tellus airborne radiometric data are available for most of the island of Ireland, the method introduced here is robust and can be utilised to develop geogenic radon potential maps with a high spatial resolution for the other parts of the island. The presented GRP map can be also used as a useful tool for land-use planning and mitigation strategies adopted by the local administration authorities.

### Study Area Description

The study area (Figure 1) includes a surface of about 4 km^2^ and is located near Graiguenamanagh, a town in County Kilkenny, Ireland, which has historic and tourist importance. The soils in the study area are mainly made of granular alluvial deposits (i.e., coarse-textured gravels and sands) of Barrow valley. The depth of the soil varies from deep soil in uphill areas to shallower depths near the shores of the Barrow river [19]. The geological setting of the study area (Figure 2) consists of Caledonian Leinster Granite which, based on the results of previous studies, is often associated with high radon concentration [20]. Moreover, the Radiological Protection Institute of Ireland (now incorporated into the EPA) identified some towns in County Kilkenny, including Graiguenamanagh, as areas that may possess high radon levels [21]. Therefore, this area was selected as the case study to firstly identify areas where indoor radon concentration is likely to be higher to aim at health risk reduction for the inhabitants and secondly, to develop a model for accurate predictions of soil gas radon based on explanatory data available at a local scale.

## 2. Materials and Methods

### 2.1. Airborne Radiometric Data

Ideally, a GRP estimation is derived from radon soil gas concentrations [22]. An alternative method to detect radon on the ground is to use airborne gamma spectrometry [18,23]. In Ireland, there are insufficient SGR data to generate useful radon potential maps. However, airborne radiometric data provided by the Tellus project are available in a high resolution for most of Ireland (http://www.tellus.ie, accessed on 30 May 2021). The airborne radiometric survey was carried out using a low-flying aircraft (flying at 60 m in rural areas and 240 m in urban areas). The data were recorded by a 256-channel gamma spectrometer (Exploranium GR820) covering 0.3–3 MeV [24]. They were integrated over flying distances of about 50 m. Potassium (^40^K), equivalent uranium (eU, estimated from ^214^Bi) and equivalent thorium (eTh, estimated from ^208^Tl) were measured in these surveys.

As shown in Figure 1, there are gaps between the airborne survey lines. To predict the radiometric data values for the points located within the gap space, the cubic spline interpolation method was utilised in QGIS software. This method is best for gently varying surfaces [25]. Using this method, raster maps of smooth surfaces of radiometric data (Figure 3, Figure 4 and Figure 5) were produced and the contour lines were generated using the Contour Extraction tool. Finally, the eU, eTh and ADR_air_ values corresponding to each SGR test point location were extracted using the Raster Sampler tool in QGIS. Note: the raw and processed data are made available in Appendix A.

### 2.2. Soil Gas Radon and Soil Gas Permeability Test

Soil gas radon activity concentrations were measured at 40 measuring sites (see Figure 1 for the location of data points, also note that the raw and processed data are made available in Appendix A) by using a Radon detector RM-2 in March 2020. Additionally, the permeability of soil was determined at each measuring point using RADON-JOK equipment. The theory of calculating permeability using the RADON-JOK instrument is derived from Darcy’s equation:·
(1)$$Q=F\cdot \left(km\right)\cdot Dp,rearrangedask=\left(Q\cdot m\right)/\left(F\cdot Dp\right)\;$$
where Q is the instantaneous flow rate, k is permeability, μ is the dynamic viscosity of the fluid, F is the friction factor and Δp is the pressure drop.

To estimate permeability at selected sites, first, the time required for pumping 2 L of air from the soil (employing negative pressure mechanically created by RADON-JOK) was recorded and depending on the time spent, the soil gas permeability was calculated. In the second step, the soil gas sample was collected from a depth of 80 cm through a hollow steel rod using a plastic 150 mL volume syringe and transferred to an evacuated ionisation chamber. After a time delay of 15 min, the chamber was introduced to the detector and the concentration of radon was measured based on the number of alpha counts that give rise to the ionisation current in the ionisation chamber [26]. Both SGR (kBq m^−3^) and permeability values (m^2^) were georeferenced and introduced into ArcGIS version 10.5 software. Spatial variability of the measured parameters was processed to generate a contour map of SGR using the ordinary kriging method, which is a geostatistical interpolator that uses randomly distributed measured data to predict values at the unsampled locations [13]. The produced contour map of the measured soil gas radon (SGR) is shown in Figure 6.

### 2.3. Distance from Fault Lines

The presence of geologic faults increases radon levels on the ground by providing favourable pathways from the source uranium-rich bedrock units to the surface. Generally, a distance less than 150 m from a major fault may have a higher influence on the anomalous increase in radon potential [27]. On the other hand, the alteration of magmatic and crystalline rocks along the non-active faults may increase the ratio of clay minerals, which decreases the permeability and the radon potential [28]. Therefore, the presence of tectonics may not essentially increase the radon potential. As can be seen in Figure 2, there is a fault structure with NW–SE orientation in the study area. To evaluate the effect of distance from a fault on soil gas radon variations, the distance of each SGR test point from the fault line was calculated for each point. To achieve this, the vectorised tectonic structure map (scale 1:100,000) (https://www.gsi.ie/, accessed on 30 May 2021) served as an input for QGIS software. Then, distance calculations were performed using the QGIS’s Nearest Neighbour Join (NNjoin) Plugin.

### 2.4. Geostatistical Model Setting and Diagnostic Tests

The map of the geogenic radon potential was elaborated using a spatial regression model, which allows for providing accurate predictions of non-stationary data on a local level. The raw soil gas radon data, as an indicator of geogenic radon potential, was considered as the response value of the model and some effective parameters (airborne radioelement concentrations of eU and eTh, air-absorbed dose rate, distance from the major fault, soil permeability and a digital terrain model) as the inputs to the model.

To evaluate relationships between predictors and the response variable, ordinary least squared (OLS) regression was used. OLS is the commonly utilised regression technique for the geostatistical modelling of radon potential [9]. It is also a starting point for all spatial regression analyses. Equation (2) presents the linear function of the repressors which can predict the response variable based on explanatory variables.
(2)$$y=b_{0}+b_{1}X_{1}+b_{2}X_{2}+...+b_{n}X_{n}+e$$
where *y* is the dependent variable predicted, *X_i_* are the explanatory variables and *b_i_* are the coefficients computed by the regression tool, representing the strength and type of relationship between *x* and *y*, and *ε* are the residuals, i.e., the unexplained portion of the dependent variables. In this study, equivalent uranium (eU), equivalent thorium (eTh), gas permeability (LogP), digital terrain model (DTM), distance from fault line (FD) and air-absorbed dose rate (ADR_air_) were considered as predictor variables for the GRP model while considering soil gas radon (SGR) concentrations as the response variable. The predictor variables in this study are selected based on the explanatory variables investigated in similar research [9,13]. To evaluate the OLS model’s validity, the variance analysis and the statistical diagnostic tests of multicollinearity (by measuring Variance Inflation Factor (VIF) [29]), heteroscedasticity (Breusch–Pagan and White tests [30,31]) and spatial autocorrelation (Durbin–Watson test [32]) were performed.

Additionally, the adjusted R^2^ value, which is a coefficient indicating how much variation in a dependent variable’s values is explained by a set of explanatory variables, has been examined. A properly specified OLS model (i.e., errors are normal, homoscedastic and independent of the repressors and the linear specification of the model is correct) should meet the following requirements [9]: (a) an adjusted R^2^ of 0.50 or higher; (b) significance of the β coefficients (*p*-values that are less than 0.05); (c) a VIF of less than 7.5; (d) a Jarque–Bera statistics for normality test (*p*-value greater than 0.10); and (e) a spatial autocorrelation test (*p*-value greater than 0.10). These statistical parameters were calculated and analysed for the proposed OLS model by running the XLSTAT add-on within Microsoft Excel [33]. Note: the results of analytical and diagnostic tests are made available in Appendix A.

## 3. Results

### 3.1. Preliminary Statistics

The descriptive statistics of measured soil gas radon (SGR) concentrations and extracted equivalent uranium (eU) and equivalent thorium (eTh) activities together with air-absorbed dose rates (ADR_air_) are reported in Table 1. The Shapiro–Wilk test [34] was also applied at a statistically significant level of 0.05 to test the normality of the distribution of radiometric data. As can be understood from Table 1, the extracted eTh values show a normal distribution; however, the eU, SGR and ADR_air_ elements did not have a normal distribution. Hence, the study area can probably be considered a natural uranium geochemical abnormal area. Furthermore, both air-absorbed dose rates and soil gas radon concentrations are directly correlated with equivalent uranium activity, which can justify the non-normality of these datasets’ distributions. For more detailed results, please see Appendix A.

SGR values ranged between 6 kBq m^−3^ and 236 kBq m^−3^ with a mean value of 88 kBq m^−3^ which is a significantly high value even for an area with a granitic bedrock. The equivalent activity of uranium and thorium ranged from 2.42 to 17.37 ppm and from 7.53 to 13.54 ppm, respectively. The calculated mean value of eU (5.39 ppm) is slightly higher than the average value of uranium in U- and Th-enriched granitic rocks (5ppm for U and 15 ppm for Th [35]). However, the mean calculated eTh (9.63 ppm) was considerably lower than the average Th content in the granite rocks. The mean air-absorbed dose rate measured for the study area is almost 2.5 times the population-weighted average absorbed dose rate of 60 nGyh^−1^ in outdoor air from terrestrial gamma radiation [36]. Comparing the contour maps of eU and air-absorbed dose rates (Figure 3 and Figure 5, respectively), it can be understood that high dose rates occurred in the areas where eU concentrations are high. This means that uranium anomalies present in the area (northern and central south sectors) may be responsible for higher dose rate values.

### 3.2. Analysis of the OLS Model

In this study, ordinary least squares regression was used to investigate the mathematical relationship between predictors and the response variable. Equation (3) represents the resulting OLS regression equation.
(3)$$\mathrm{SGR}\;(kBq\cdot m^{-3})\;=78.81-0.15\;(DTM)\;-\;2.87\;E-02\;(DF)\;+\;12.53\;(eU)\;+\;0.53\;({\mathrm{ADR}}_{\mathrm{air}})\;+\;11.31\;(eTh)\;+\;18.39\;(Log\;P)$$
where SGR = soil gas radon concentration (kBq m^−3^), ADR_air_ = air-absorbed dose rate (nGyh^−1^), Log P = permeability of the soil (m^2^), eTh = equivalent content of thorium (ppm), DTM = digital terrain model (m a.s.l.), DF = distance from the major fault (m) and eU = equivalent content of uranium (ppm).

To avoid using variables that may cause instability in the model, multicollinearity was tested through the Variance Inflation Factor (VIF) statistic before setting the regression model. As shown in Table 2, the VIF value did not exceed the recommended value of 7.5 for any of the explanatory variables. The result of the regression model analysis (Table 2) states that the variables eU, eTh and Log P bring a significant amount of information to explain the variability in the dependent variable (SGR). However, the variables DTM, DF and ADR_air_ are not consistently significant. Figure 7 shows the chart of standardised regression coefficients of the model. The chart allows us to directly compare the relative influence of the explanatory variables on the dependent variable and their significance. The higher the absolute value of a coefficient, the more important the relative influence of a variable is. It can be concluded that among the significant variables, Log P and eU are the most influential ones. This can be explained by the fact that these two parameters mainly control the source of radon (as uranium is the parent of radon) and also the mobility of the gas toward the surface, therefore having influence was anticipated for them. For more detailed results, please see Appendix A.

### 3.3. Validity of the Model

Table 3 summarises the results of diagnostic tests performed for the proposed OLS regression model. Given the *p*-value (<0.0001) of the F statistic computed by Fisher’s test, and given the significance level of 5%, the information brought by the explanatory variables is significantly better than what a basic mean would bring. Given the adjusted R^2^, it can be concluded that the proposed model explains approximately 60% of the variation in the dependent variable (SGR). The chart in Figure 8 visualises the correlation between the measured SGR and predicted values, the regression line (the fitted model R^2^ = 0.66), and two 95% confidence intervals (i.e., given the assumptions of the linear regression model, residuals should be normally distributed, meaning that 95% of the residuals should be in the interval (−1.96, 1.96)). All values outside this interval are potential outliers. As can be seen in Figure 8, no potential outlier value was identified among the measured SGR values. Moreover, the Jarque–Bera statistics value is not significant, which indicates that the residuals follow a normal distribution.

The Durbin–Watson test is a measure of autocorrelation (also called serial correlation) in residuals from regression analysis; the null hypothesis is no spatial correlation. As the computed *p*-value for this test is greater than the significance level alpha = 0.05, one cannot reject the null hypothesis. Furthermore, the Breusch–Pagan and White tests are used to test heteroscedasticity. As the calculated *p*-values of these two tests are greater than the significance level alpha = 0.05, it can be concluded that residuals are homoscedastic. The above-mentioned diagnostic test results demonstrate that the basic assumptions underlying the OLS regression analysis have not been violated. For more detailed results, please see Appendix A.

### 3.4. Geogenic Radon Potential Mapping

In the previous sections, the relationship between radon-related variables and the soil gas radon concentration was obtained. The geogenic radon potential (i.e., the quantity of radon directly related to the local geology) is defined as the soil gas radon values predicted based on the regression equation which calculates SGR solely based on predictor variables. To produce the map of the spatial distribution of GRP for the study area, the empirical Bayesian kriging (EBK), an automatic geostatistical interpolator, was used in ArcGIS version 10.5 software. This tool creates prediction surfaces based on restricted maximum likelihood estimation. Additionally, it allows for analysis of the uncertainty in the semivariogram model by a process of data sub-setting and simulation to estimate a range of semivariogram models [37].

In this study, the EBK was employed for three purposes: (a) for cross-validation of the results, (b) to create a GRP prediction surface and (c) to estimate the error of predicted values. Table 4 shows the results of cross-validation of the predicted values (i.e., the geogenic radon potential) and measured SGR activities. According to this table, the mean prediction error (ME = 1.98) and the mean standardised error (MSE = 0.035) are close to zero, indicating that the interpolation method is unbiased (centred on the true values) and the model is accurate; the average standard error (SE = 50.68) is higher than the root mean squared prediction error (RMSE = 48.38), suggesting that the interpolation method slightly overestimates the variability in the predictions [13].

The GRP surface predicted based on the soil gas radon concentrations is shown in Figure 9, along with its corresponding predicted standard error surface, which illustrates the amount of uncertainty associated with the predicted SGR value. As shown in Figure 9, high GRP values (>85 kBq m^−3^) can be found in the eastern and northwestern sectors of the study area. Just for the southwestern part, the GRP is in the low range (50 kBq m^−3^) and the remaining parts are characterised by medium GRP levels (50 to 85 kBq m^−3^). The uncertainty in GRP prediction shown in Figure 10 is generally in the low to medium range (less than 35 kBq m^−3^); however, for areas along the borders of the study area, rather high values (up to 52 kBq m^−3^) can be predicted. A possible source of error in those areas would be an insufficient number of SGR test points that occurred due to sampling limitations (i.e., presence of saturated soil and improper site access).

### 3.5. Comparison of Predicted Radon Potentials with Neznal’s Radon Index

To better understand and further investigate the geogenic radon potential of the study area, the radon potential predicted based on spatial regression in the previous section was compared with a radon index (RI) calculated based on the Neznal method [26]. The radon index [26] was formulated as follows:(4)$$\mathrm{RP}\;=\;\left(\mathrm{CSGR}-1\right)/\left(-\log\;k\;-10\right)$$
where the unit of CSGR (soil gas radon concentration) is (KBq m^−3^). K (m^2^) is the permeability of the soil. The calculated RP (radon potential) enables the determination of RI as low, medium or high (if RP < 10, then RI is low; if 10 ≤ RP < 35, then RI is medium; if 35 ≤ RP, then RI is high). A summary of the results of radon potential categorisation based on the Neznal method is presented in Table 5.

According to the histogram of the Neznal categorisation index (Table 5 and Figure 11), only 7.5% of calculated RIs show a low radon index category, 32.5% at the medium range and 60.0% at the high level. This confirms that the area under investigation is a high-radon-risk area.

Figure 12 shows the distribution of calculated radon index values produced using the nearest neighbour interpolation method. Similar to the map of predicted radon potential in Figure 9, areas with high geogenic radon risk are located within the northwest and southeastern confines of the study area. Although among the scientific society, Neznal’s method has been proven to be an acceptable approach to formulate the radon potential, it is based on soil gas radon concentrations and permeability alone. Here, we apply a well-recognised and robust spatial regression model where a set of effective parameters are introduced as inputs to map radon potential [13]. We see the two approaches as complementary in nature, with the advantage that the spatial regression model includes a greater number of variables to define areas of geogenic radon potential.

## 4. Discussion

A set of radiometric data and geogenic parameters together with soil gas radon and permeability measurements were used to model the geogenic radon potential of a granitic area. The proposed model was successful in explaining 60 percent of the variations in the soil gas radon. It is suggested that considering other radon-affecting factors in the OLS model would help to increase the goodness of the prediction. Other radon-affecting factors mainly include soil type, grain size, water content and porosity [38]. In this research, due to the lack of information about additional parameters (at least on a local scale), we were not able to fully consider their effects.

Rainfall and precipitation are among the important events that can change near-surface environmental gamma rates by affecting radon progenies ^214^Pb and ^214^Bi [39]. Furthermore, water accumulation in the surface soil may act as a shield and prevents radon emission. The accumulation of radon in a stable boundary layer can also have a considerable influence on gamma levels [40]. Moreover, hydrological movement at sites with high radon fluxes may represent a source of false alarms regarding radioactivity levels [41]. As mentioned in Section 2.1, the equivalent uranium concentrations measured in the frame of the Tellus project are estimated from the ^214^Bi photoelectric peak. Knowing the fact that rainfall events, precipitation or other meteorological parameters can highly affect the detection results of airborne gamma-ray spectrometry, and also considering that Ireland is a country in which rainfall and precipitation are frequent and soils can be saturated during wet seasons, it would be necessary to perform radon background correction for airborne gamma-ray spectrometry results to improve the accuracy of measurement and avoid false estimates of radon anomalies [42].

## 5. Conclusions

Tellus airborne radiometric data provide valuable information about natural radioactivity (gamma radiation) of near-surface rocks and soils using a gamma-ray spectrometer. The data have many applications in environmental monitoring and geological mapping. As an example, the estimated natural radionuclide concentration of uranium, thorium and also the air-absorbed dose rate can be used as an indicator of the presence of a radon (including thoron) source. In this research, to prepare a geogenic radon potential map, radiometric data were used in combination with geogenic factors which account for radon mobility and transport from the source to the ground level. Airborne radioelement concentrations (i.e., eU and eTh), air-absorbed dose rate, distance from the major fault, soil permeability and a digital terrain model were assumed as the main predictors and soil gas radon was considered as the response value of an ordinary least squares regression model. Through diagnostic tests, the validity of the developed model was evaluated. At the later stage, empirical Bayesian kriging, an interpolator tool, was used to produce the geogenic radon potential maps and estimate the uncertainty of predictions. The method introduced here can be considered a promising tool to produce GRP maps for other regions, especially in Ireland. Moreover, it is possible to improve the model by considering the effect of additional parameters (e.g., water content of the soil) for which there are valuable literature data in Irish databases. For this research, it was not possible to integrate this information as the selected study area was relatively small, making it incompatible with the resolution of the data. Finally, according to the standardised regression coefficients of the explanatory variables (Figure 7), permeability was found to be the second most effective parameter in predicting soil gas radon activity; hence, it strongly affects GRP levels. As observed here, the soil permeability can be very spatially variable even for intervals of tens of meters. Therefore, it is suggested to use data on in situ permeability measurements rather than indirect estimations for radon mapping purposes.

## Figures and Tables

**Figure 1 ijerph-19-15910-f001:**
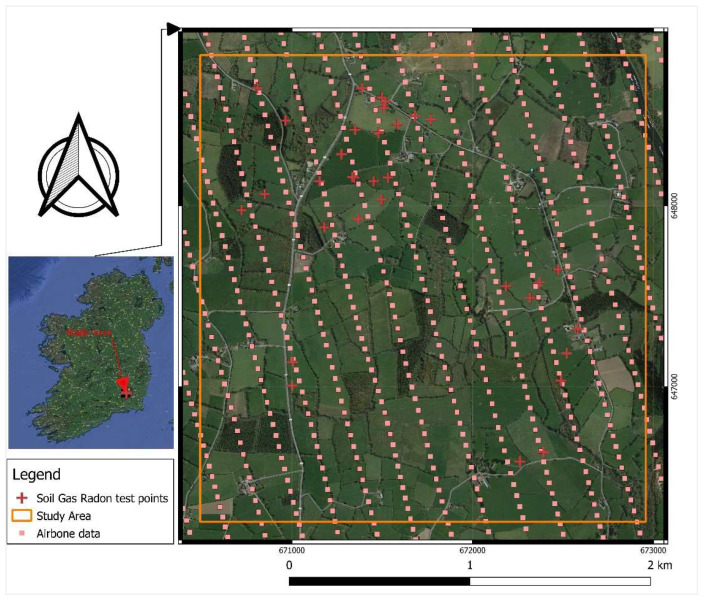
Satellite image of the study area with the location of the soil gas radon testing and airborne radiometric survey points.

**Figure 2 ijerph-19-15910-f002:**
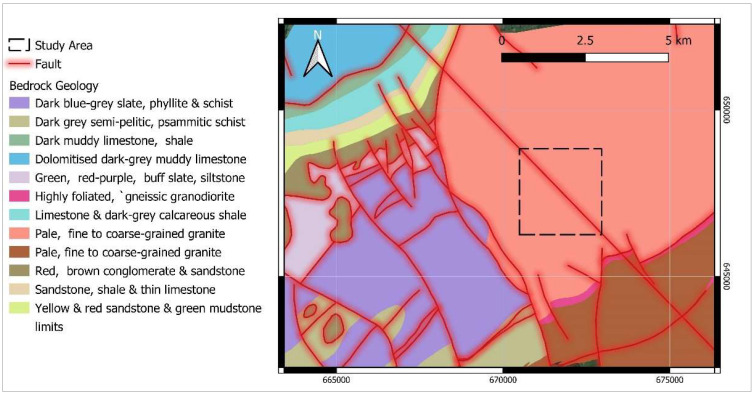
Geological map of the study area, extrapolated from Bedrock Geology map (scale 1:100,000) of Geological Survey Ireland (GSI).

**Figure 3 ijerph-19-15910-f003:**
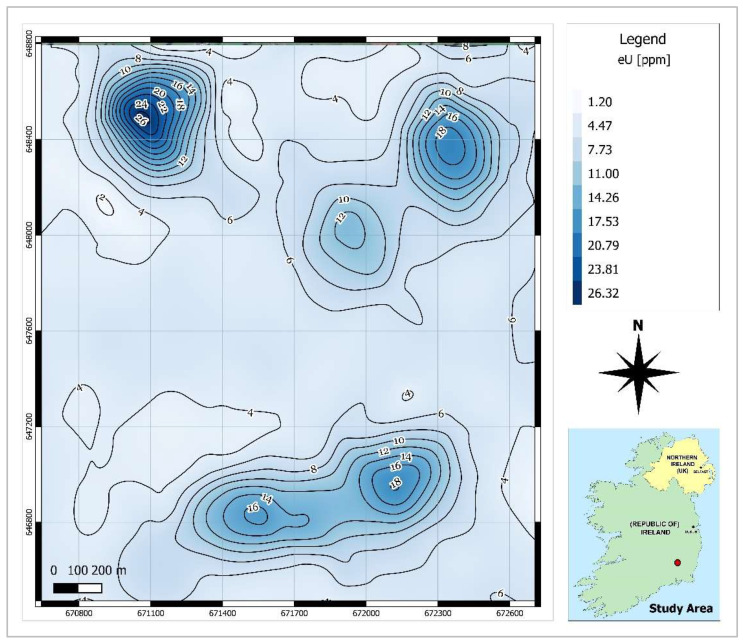
Contour map of equivalent uranium concentration (eU).

**Figure 4 ijerph-19-15910-f004:**
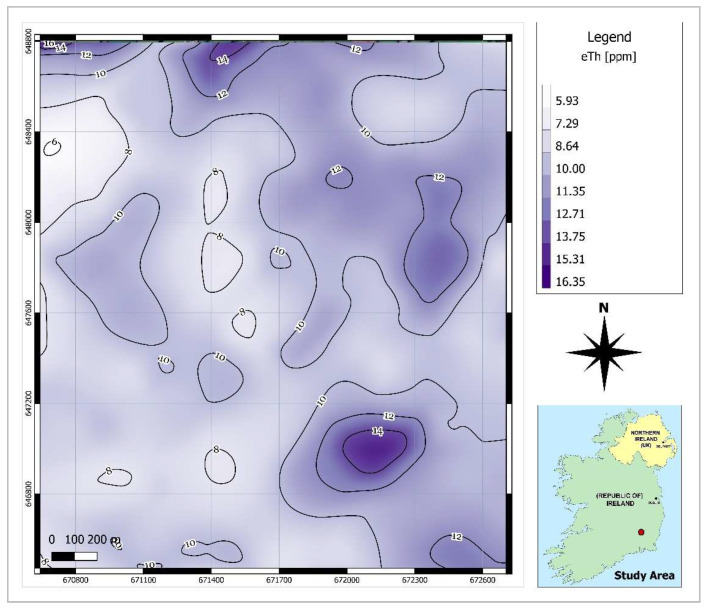
Contour map of equivalent thorium concentration (eTh).

**Figure 5 ijerph-19-15910-f005:**
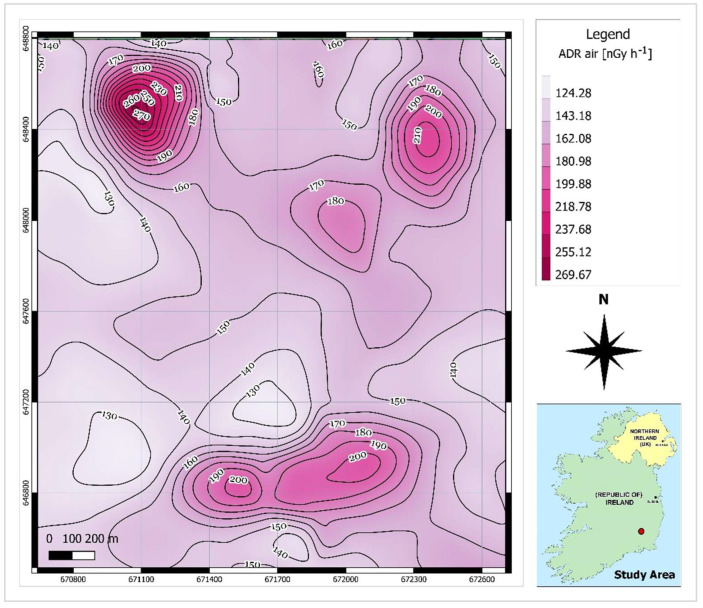
Contour map of air-absorbed dose rates (ADRair).

**Figure 6 ijerph-19-15910-f006:**
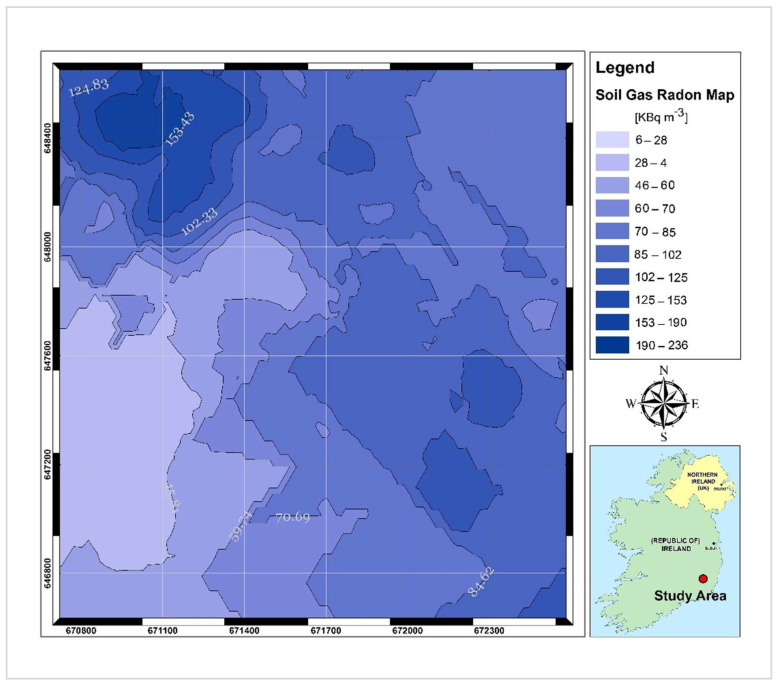
Contour map of measured soil gas radon (SGR) activities.

**Figure 7 ijerph-19-15910-f007:**
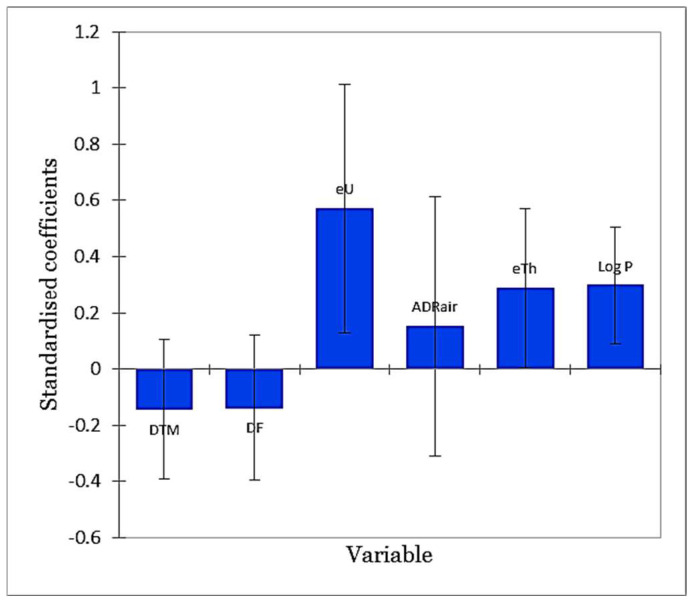
The standardised regression coefficients of the explanatory variables (95% confidence interval).

**Figure 8 ijerph-19-15910-f008:**
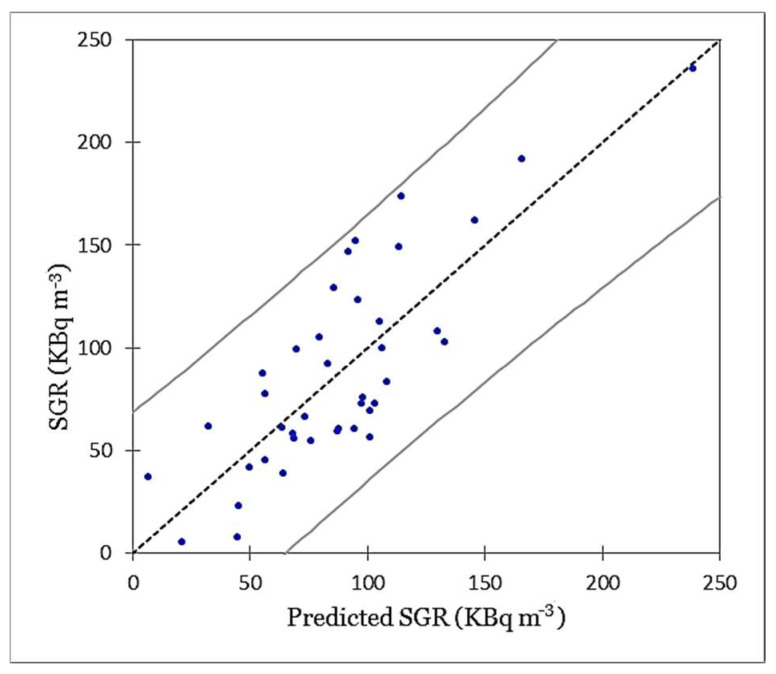
Predicted SGR values versus the measured ones (95% confidence intervals).

**Figure 9 ijerph-19-15910-f009:**
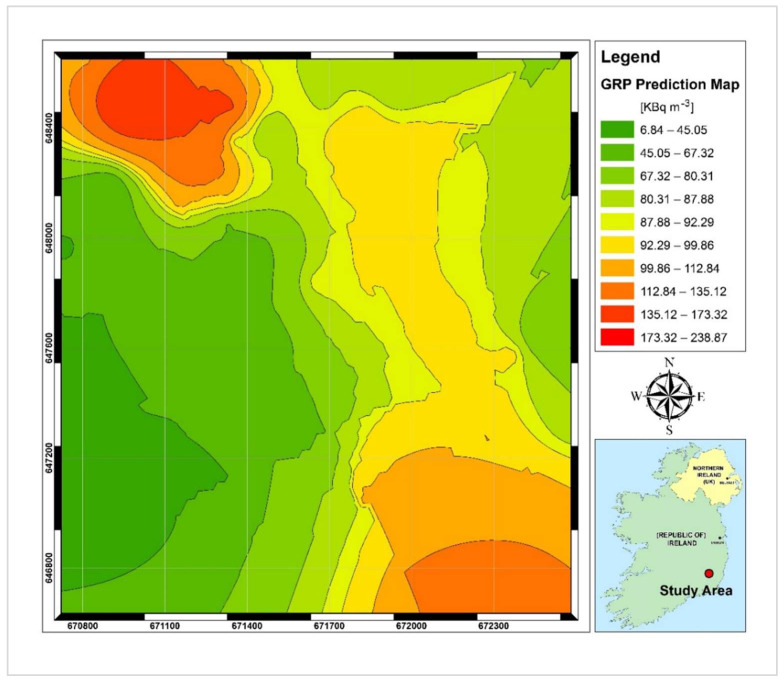
Predicted Geogenic Radon (GRP) concentrations.

**Figure 10 ijerph-19-15910-f010:**
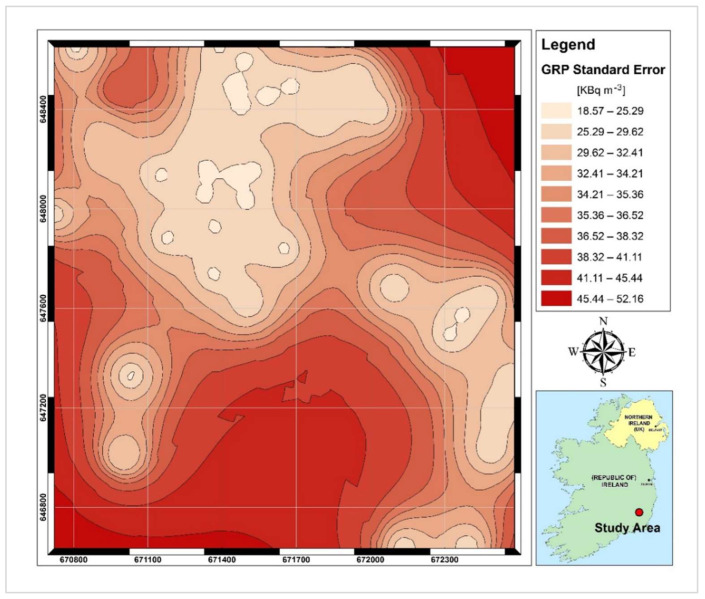
Map of GRP prediction error.

**Figure 11 ijerph-19-15910-f011:**
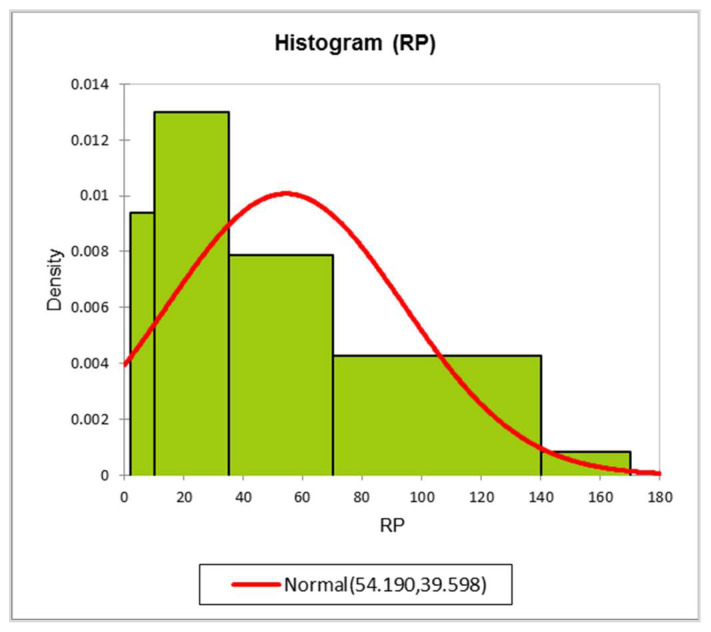
Histogram of Neznal categorisation indexes. Redline shows the normal distribution.

**Figure 12 ijerph-19-15910-f012:**
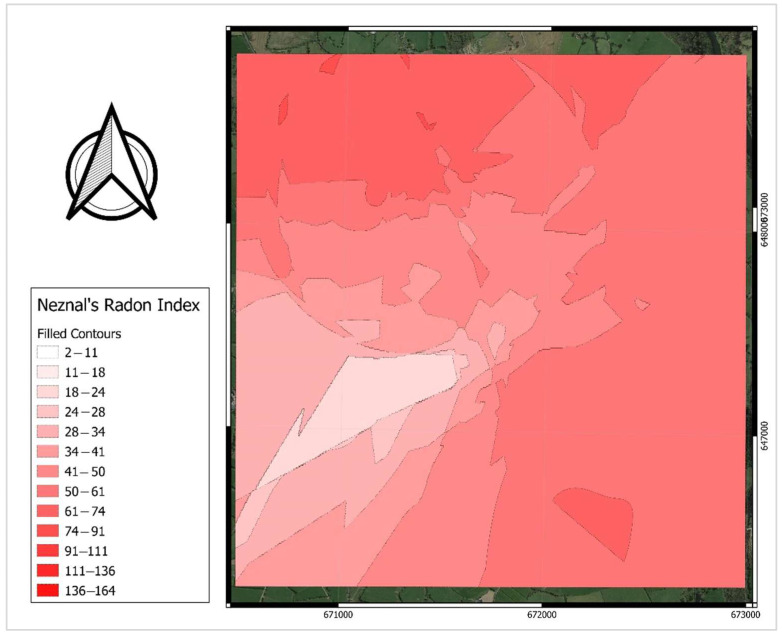
Distribution map of Neznal’s radon index.

**Table 1 ijerph-19-15910-t001:** Descriptive statistics of extracted airborne radiometric data and soil gas radon concentrations.

Statistic	eU (ppm)	eTh (ppm)	ADR_air_ (nGy h^−1^)	SGR (kBq m^−3^)
Number of data	40			
Minimum	2.42	7.53	124.81	5.60
Maximum	17.37	13.54	214.45	236.00
1st Quartile	4.44	8.60	150.62	57.90
Median	4.94	9.61	154.65	74.30
3rd Quartile	5.47	10.38	159.83	109.25
Mean	5.39	9.63	154.70	87.96
Standard deviation	2.25	1.26	14.14	49.34
Skewness (Pearson)	3.78	0.64	1.50	0.88
Kurtosis (Pearson)	17.44	0.54	6.29	0.64
Geometric mean	5.10	9.55	154.09	71.95
Geometric standard deviation	1.36	1.14	1.09	2.10
Shapiro–Wilk test (W)	0.62	0.96	0.85	0.94
	Reject normality (*p* < 0.0001)	Normal distribution (*p* = 0.22)	Reject normality (*p* < 0.0001)	Reject normality (*p* = 0.04)

**Table 2 ijerph-19-15910-t002:** Main statistics of the coefficients of the explanatory variables for the OLS model.

Source	Coefficient	SE	t	Pr > |t|	VIF
Intercept	78.81	131.03	0.60	0.55	-
DTM	−0.15	0.13	−1.18	0.25	1.44
DF	−0.03	0.03	−1.09	0.28	1.55
eU	12.53	4.77	2.63	**0.01 ***	4.60
ADR_air_	0.53	0.79	0.67	0.51	5.01
eTh	11.31	5.46	2.07	**0.05 ***	1.88
Log P	18.39	6.32	2.91	**0.01 ***	1.02

* Statistically significant.

**Table 3 ijerph-19-15910-t003:** Descriptive statistics of extracted airborne radiometric.

Parameter	Value	*p*-Value
Observations	40	
R^2^	0.66	
Adjusted R^2^	0.60	
AICs	282.63	
Fisher’s F test	10.72 (DoF = 6)	<0.0001
Durbin–Watson test (DW)	2.31	0.50
Breusch–Pagan test (LM)	5.95	0.43
White test (LM)	26.97	0.47
Jarque–Bera test	3.06	0.22

**Table 4 ijerph-19-15910-t004:** Discus: Results of executing Cross-validation using Empirical Bayesian Kriging.

Parameter	Value
Count	40
Mean Error	1.98
Root Mean Square Error	48.38
Average Standard Error	50.69
Mean Standardised Error	0.035
Root Mean Square Standardised Error	0.96

**Table 5 ijerph-19-15910-t005:** Summary of the results of radon potential categorisation based on the Neznal method.

Summary Statistics						
Variable		Observations	Minimum	Maximum	Mean	Std. Deviation
RP		40	2.21	166.63	54.19	39.56
**Descriptive Statistics for the Intervals (RP)**						
RI	RP		Frequency	Relative frequency	Density (Data)	Density (Distribution)
	Lower bound	Upper bound				
Low	2	10	3	0.075	0.009	0.038
Medium	10	35	13	0.325	0.013	0.182
High	35	70	11	0.275	0.008	0.341
	70	140	12	0.300	0.004	0.330
	140	170	1	0.025	0.001	0.013

## Data Availability

Not applicable.

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
