# Peer review of "Detailed Geogenic Radon Potential Mapping Using Geospatial Analysis of Multiple Geo-Variables—A Case Study from a High-Risk Area in SE Ireland"

_ijerph, 2022, doi:10.3390/ijerph192315910_

Round 1
Reviewer 1 Report
The aim of the article titled „Detailed geogenic radon potential mapping using geospatial analyses of multiple geo-variables. A case study from a high-risk area in SE Irland” was an investigation of GRP (Geogenic Radon Potential) via performing spatial regression analysis on radon-related variables such as radionuclides contents, air absorbed dose rate, soil permeability, distance from major faults.
Major remarks
On what basis did you choose the location of sites where radon concentrations in soil were measured?
Were all measurements such as soil gas radon concentration and soil permeability performed during similar atmospheric conditions?
Dear Authors, why did you not decide to calculate GRP Geogenic Radon Potential on the basis of Soil Gas Radon Concentrations (SGR) and soil permeability using the equation proposed by Neznal, 2004, and then compare the calculated GRP with GRP map?
How did you predict the value of Geogenic Radon Potential GRP on the basis of SGR Soil Gas Radon assessed using Equation 3? Are those the same data but differently named once as SGR and once as GRP? If yes, you should rather compare the values of assessed SGR = GRP with GRP calculated using the Neznal equation (GRP = CRn / (−log10 (k) − 10), where k (m2) is the soil permeability and CRn is the concentration of radon (kBqm−3) in the soil-gas)
Line 174-175
“To evaluate relationships between predictors and the response variable, Ordinary Least Squared (OLS) regression was used”.
Ordinary Least Squared (OLS) regression should be used when the data distribution is normal. According to Table 1, eU, ADRair, and SGR data do not have normal distributions, which means that other statistical tests should be applied.
Line 190-191
To evaluate the OLS model validity, the variance analysis and the statistical diagnostic tests of Multicollinearity (by measuring 191 Variance Inflation Factor (VIF) [28]))(…..)
Variance should be analysed when data have a normal distribution.
Minor remarks:
Line 36-37
“The main source of radon is the gas flux from soils and rocks that host uranium and radium-bearing minerals”
The main source of radon (both isotopes Rn-220 and Rn-222) in the gas flux is radium or uranium and thorium. In the soil gas the Rn-220 concentrations can be even higher than Rn-222.
The main source of radon in the buildings is the gas flux from soils and rocks that host uranium and radium bearing minerals.
Line 49-50
“Radon mapping can be mainly performed in two methods: a) directly, based on spatial distribution indoor radon measurements and b) indirect estimation of radon potential based on geogenic information such as radiometric surveys (terrestrial or airborne), geochemical data (e.g. U and Ra content), soil gas radon measurements, soil permeability, faults, etc.”
Radon mapping based on the results of soil gas radon and soil permeability measurements used to calculate geogenic radon potential GRP is not indirect.
Figure 1
According to the Reviewer, the fault should not be indicated as a dotted line.
Line 373-374
“Finally, the results of this study show that soil gas permeability is an important parameter that strongly affects GRP levels”
Please, explain which part of your research indicates that gas permeability strongly affects GRP level?
Consider comparing assessed SGR which is then treated as GRP (as I suppose) with calculated data of GRP following the Neznal (2004) equation. Geogenic radon potential is not equal to soil gas radon concentration but the soil permeability should be included.
Author Response
We thank the reviewer for his/her careful thoughts and constructive suggestion to improve our paper. We tried to consider his/her comments as much as possible and modify the text and one of the figures to address the reviewer’s concerns. With regard to point-by-point responses to his/her comments, please see the attached file.

Reviewer 2 Report
The authors presented an interesting research on Radon Potential Mapping (RPM). The manuscript is written and proceeded well. I have only few following concerns:
1. The study area under consideration is very small (4 Km2). Why such a compact region is selected?
2. RPM is generally estimated from the soil radon content. But it was not determined in this study, which could have been done easily.
3. The basis of the results obtained in this manuscript is previously published/obtained data of air-borne gamma spectroscopic analysis of the study area. The detector position in these measurements were around at 60 m and 240m. The accuracy and correctness of the RPM from the data obtained at 240 m is questionable.
4. The authors may also add the methodology, how Radon potential was obtained from the air borne measurements of Uranium, Thorium and Potassium along with the error analysis, if possible.
Author Response
We truly thank reviewer#2 for his/her positive feedback. The reviewer well-understood the main goal of our manuscript and the implications of our method. Therefore, we carefully addressed the reviewer’s concerns in the revised version and the required modifications were made to improve the manuscript. Please see the attached file for point-by-point responses to the reviewer's comments.

Round 2
Reviewer 1 Report
no comment